# A 3D Printing Method of Cement-Based FGM Composites Containing Granulated Cork, Polypropylene Fibres, and a Polyethylene Net Interlayer

**DOI:** 10.3390/ma16124235

**Published:** 2023-06-07

**Authors:** Daniel Pietras, Wojciech Zbyszyński, Tomasz Sadowski

**Affiliations:** 1Department of Solid Mechanics, Lublin University of Technology, Nadbystrzycka 40 Str., 20-618 Lublin, Poland; d.pietras@pollub.pl; 2Civil Engineering Laboratory, Lublin University of Technology, Nadbystrzycka 40 Str., 20-618 Lublin, Poland; w.zbyszynski@pollub.pl

**Keywords:** 3D printing, concrete printing, additive manufacturing, continuous reinforcement, fibre reinforcement, cork

## Abstract

The increasing popularity of additive manufacturing technologies in the prototyping and building industry requires the application of novel, improved composite materials. In this paper, we propose the use of a novel 3D printing cement-based composite material with natural, granulated cork, and additional reinforcement using a continuous polyethylene interlayer net combined with polypropylene fibre reinforcement. Our assessment of different physical and mechanical properties of the used materials during the 3D printing process and after curing verified the applicability of the new composite. The composite exhibited orthotropic properties, and the compressive toughness in the direction of layer stacking was lower than that perpendicular to it, by 29.8% without net reinforcement, 42.6% with net reinforcement, and 42.9% with net reinforcement and an additional freeze–thaw test. The use of the polymer net as a continuous reinforcement led to decreased compressive toughness, lowering it on average by 38.5% for the stacking direction and 23.8% perpendicular to the stacking direction. However, the net reinforcement additionally lowered slumping and elephant’s foot effects. Moreover, the net reinforcement added residual strength, which allowed for the continuous use of the composite material after the failure of the brittle material. Data obtained during the process can be used for further development and improvement of 3D-printable building materials.

## 1. Introduction

Additive manufacturing is a type of manufacturing using special layer-laying techniques, allowing for the production of different shapes and forms [1]. The main idea behind it is to reduce human labour and increase precision while also reducing waste [2,3,4,5]. There have been many attempts of introducing additive manufacturing, or so-called 3D printing (3DP), into the scientific and engineering world, and each used a different approach. Besides changing layer-laying techniques, there have been attempts of using a large variety of materials [2,3,5,6,7]. While materials such as polymers promptly gained popularity because of their highly plastic behaviour and ease of forming, more rigid materials such as concrete, mortar, or clay were not in favour [3,6,8,9]. Notably, 3DP technologies brought a new level of freedom of form and shape into the building industry [4,5,10,11]. The lack of need for external formwork not only allows designers to change the design of certain elements to a more sophisticated and complicated one, but it also cuts costs by up to 54%. It is estimated that 34% of the cost of rising monolithic construction results from the formwork itself, and an additional 20% is generated by the work needed for its use and maintenance [2,7,12]. The use of programmed machines allows for excluding man-made errors and continuous operation in a building site [2,3,6,10].

Although there are a few 3DP methods [13], and quite a substantial number of material mixtures are used in different methods and different ideas of reinforcement, there is still not enough information about how printed materials behave, and what possibilities of improvement there are [3,5,10,14]. Each method of printing has its flaws. Binder-jetting and D-shape printing techniques, while offering the huge advantage of free forming, need after-treatment, which takes time and a lot of additional labour [15,16]. Contour crafting (CC) and concrete printing (CP), while giving good results in buildability, have problems with maintaining overhangs and elements such as lintels, which have to be added separately [2,17]. What is important is that most of the materials used in 3D printing have inferior thermal capabilities, which limits the use of this technology in colder and more harsh environments. Besides changing the structure of elements so that they obtain better thermal capabilities [18], there are ideas of using different materials to insulate the structures by adding them as a layer inside of elements or changing the thermal behaviour of the main printing material by changing its mixture composition [3,19]. The addition of foam insulation inside 3D-printed concrete elements might work as in the case of sandwich composites, where structures with concrete outer shells and foam cores will be lighter than monolithic concrete structures while still maintaining considerable strength and additional thermal properties [20,21]. Materials can be also designed to be functionally graded (FG), which means that the material’s properties change stepwise in a sequence of layers or according to desired gradient [21,22]. To increase the building capabilities of CP, there have been attempts of adding reinforcement to printed layers via the addition of fibre reinforcement [13,23] into the material mixture, the addition of continuous reinforcement between the printed layers [2,3,24], or even with the use of nails [25].

Even though there are mixtures used for 3D printing, which should be able to withstand 80 MPa in uniaxial compression tests after curing, most of them are mortars or low-grain concrete mixtures rather than actual concrete mixtures [26,27,28]. In other words, most of the mixtures used in CP technology do not contain coarse aggregates, and therefore they do not count as regular concrete mixtures. The use of coarse aggregates is problematic in CP because of the risk of jamming the machine during the process and the high risk of a structure being uneven, which might lower the strength of the bond between layers [5]. The use of low-density materials such as cork granules, which are mostly by-product waste, as additional filler might be the solution to this problem [29,30]. Due to the porous structure of natural cork, not only is it light but it also has good thermal properties, which might positively impact the final mixture’s properties [31,32,33]. Therefore, the current important problem is to explore highly renewable and ecological materials such as cork, not only to improve the materials used in the building industry but also to make 3DP technology more viable and available for future generations.

Moreover, 3DP technology should be continuously optimised. Different ways of mixture extrusion might have an impact on printing times and the precision of the final product [2,6]. The influence of changes in the concrete mixtures such as additives and admixtures, water-to-cement ratio, the use of different aggregates, and fibre reinforcement can have a serious impact on the behaviour of printed elements [15,34,35,36]. Because of the well-known facts of the positive impact of continuous reinforcement on concrete structures, continuous reinforcement in 3D-printable forms should be taken into consideration as an inseparable part of the whole process [5,7,37].

Therefore, in this study, we explore the properties of a new CP method containing three additional phases. Besides the use of novel extrusion machines based on the ideas of popular CP machines, a 3D-printable composite material containing cork granules as aggregate and two types of reinforcement in the form of polypropylene fibres and a continuous polyethylene net were used [7,17,38]. The main mixture of CP composite was based on a few earlier studies and was adjusted for the capabilities of available equipment.

## 2. Materials

The final mixture was prepared using materials such as cement, sand, granulated cork, water, additives, and reinforcements. The amount of every material was adjusted to achieve the mixed properties necessary for the extrusion of the material. Properties such as extrudability, flowability, buildability, and open time were taken into account. It should be stated that we initially used gravel with a maximum dimension of 4 mm as the main source of coarse aggregate, but due to its properties leading to the reoccurring high risk of machine jams, the use of gravel was discontinued.

EN 197-1 CEM I 42,5 R Portland cement was used as the matrix. Silica sand was used as the main source of fine aggregates. Granulated cork with a maximum diameter of 3 mm was used as a lightweight aggregate. Cork was chosen because of its good thermal properties and low density, as stated in [19]. The used aggregates and the grading curves of their composition are shown in Table 1. Tap water was used as a source of batched water. Figure 1 shows a comparison of the aggregates and their composition’s grading curves.

Plasticising and aerating admixtures were added to lower the need for the use of water in the proposed mix and increase its frost resistance. The fluid admixture should be added to water in batches, amounting to 0.3% to 0.6% of the mass of added cement. The properties of the admixture declared by the manufacturer are shown in Table 2.

Fib-34, class 1a, and 12 mm polypropylene fibres were used to minimise the influence of contraction during curing and to facilitate the stabilisation of the mix throughout the whole process of specimen preparation. According to the manufacturer, 0.003% to 0.006% of fibres shall be added for every kilogram of cement in the mixture. The number of fibres should be adjusted according to needs and equipment capabilities.

A polyethylene monofilament net with a 7 × 7 mm size of mesh was used as the source of main reinforcement. No additional coatings were applied on the net before its implementation in the structure of the composite. Besides being weather- and water-resistant, polyethylene has low reactivity, which allows for its use in different aggressive environments. In addition to that, its non-magnetic nature enables its use in places where magnetic fields might produce risks for the use of standard reinforcement. Because of the high popularity of the material itself, it should not be a problem to manufacture nets suitable for special use in 3D printing. The use of nets as reinforcement in 3D-printed elements can lower slumping effects and minimise the so-called elephant’s foot effect of lower layers. A mix of sand and granulated cork in a specific ratio was prepared to cover the need for aggregates. During the process of obtaining the final mixture, gravel use was discontinued because of its capability to jam the machinery. Part of the mixture that should be covered by gravel was substituted by sand in an equal amount. The water-to-binder ratio of the mixture was set at 0.48. This ratio is not far off the commonly proposed for 3D-printable mixtures [7,17,21,27,39]. Although the ratios are different from commonly used ones, the water-to-cement ratio used in this mixture allows for the easy extrusion of material, in addition, it should be stated that a lower ratio is preferable and more beneficial for 3DP processes. The final composition is shown in Table 3.

Although the grading curve of the composition of aggregates does not perfectly match the lower and upper limits presented by PN-B-06712:1997, [40], having a high number of particles with a size of 0.25 mm or larger and fewer particles smaller than that size substantially lowers the water demand of the mix. The use of aggregates smaller than 4 mm allows for the use of the mixture in the extrusion process.

A special mixing procedure was carried out before the extrusion of the material. All of the components have to be properly weighed before further steps. After the preparation of the components for the needed amount of mixture, one should proceed with a preliminary mix of dry components. Cement, sand, cork granules, and polyethylene fibres should be mixed until uniform coverage of all the ingredients with cement. For better fibre distribution in the mix, it is suggested to add the first half of the prepared fibres at the beginning of dry mixing and the second half, halfway through. In the next step, half of the prepared water and all of the admixture should be added to the dry mix and mixed mechanically for about 2 min until components are spread uniformly, and consistency of the mixture is achieved. After 2 min of mixing, even if the expected effect is not achieved, the rest of the water should be added, and all of the mixtures should be mixed for 3 min clockwise and 3 min in the opposite direction. The effects of longer mixing times are not known, but it has been established that mixing times lower than a total of 6 min significantly worsen the spread of mixture components, which leads to imperfections in the fresh mixture and this in turn leads to imperfections in cured elements.

Fresh mixtures have to be subjected to preparation time, which consists of leaving the fresh mixture for 20 min at room temperature and humidity. After the first 10 min of preparation time, the fresh mixture should be stirred thoroughly. The mixing material that stayed on the walls of the container with stirred material should be taken into consideration. When the point of 20 min of preparation time is reached, the fresh mixture should be stirred again and used for specimen preparation. The processed mixture is usable for about 80 min, and after that period, it is prone to cause extrusion system blockage.

## 3. Methods

### 3.1. Specimen Preparation

#### 3.1.1. Forming

Specimen formation began at the end of the preparation time. The fresh mixture was formed into shape using special machinery designed for material extrusion. The machine consists of three main elements and an additional double-axial control system, which allows for the control of the length and height of the printed elements. The first element called the hopper allows for the storage of the material and continuous operation of the machine. The fresh mixture was manually delivered into the hopper’s chamber and was then transferred into the printing nozzle using a powered spiral screw conveyor. The printing nozzle was a rectangular steel tube with a mounting bracket, allowing for its use in the machine. A sliding table mounted on plain bearings was prepared for the length control of the printed specimens. Height control was achieved by the use of two vertical plain bearings, which stabilised the axis and were adjusted using a powered screw-type manipulator.

The specimen-forming technique consisted of four main steps: mixture laying, reinforcement implementation, height adjustment, and length axis reset. The steps can be repeated as many times as needed. The reinforcement was applied in the interface region by placing the net on top of the layer, without the use of any additional external forces. The net was placed after each layer of the cement-based mixture was formed. The placement of the reinforcing net and its exemplary model is visualised in Figure 2. The formed specimens were stored and cured in preferred conditions. Four specimens were prepared without using the net reinforcement to measure the influence of reinforcement on the specimens’ mechanical behaviour.

#### 3.1.2. Curing

Freshly formed specimens were cured at room temperature ranging from 21 to 25 °C and air humidity of about 40 to 60%. The specimens were not painted or sprayed in any additional water-retaining formulas, nor were they treated with water by any means. The curing conditions were chosen because of the nature of in situ 3D printing processes, which hinders the possibility of watering the mortar during the time of curing. In terms of assumed circumstances, for bulk elements, where no formwork is used, typical curing underwater would be problematic [41]. Moreover, due to the complicated design of 3D-printed elements, the use of additional surface-applied water-retaining formulas or frequent and even water spraying might not be possible. All of the specimens were cured for 28 days before testing.

#### 3.1.3. Machining

The cured specimens were prepared for tests via additional machining, which allowed for better fixtures in testing apparatuses. An ATM Brillant 250 wet abrasive cut-off machine was used for cutting flat surfaces and dividing long specimens into smaller cubical pieces for compression testing. The machine provided high-quality flat, parallel surfaces, which were crucial for obtaining accurate results. Cutting was carried out using friction cutting discs with a cutting speed of 90 mm per minute. This speed was chosen because of no change in quality for lower cutting speeds. A Luna MLF 1022 universal drilling and milling machine was used for upper-face filing. All of the tested specimens were filed using a rotating file, which allowed for levelling the upper layer to maintain a similar height throughout the specimen. The machine tolerances of ±0.1 mm allowed for maintaining parallelism and straightness of the bottom and upper surface within the tolerances required in PN-EN 12390-1:2021, [42].

#### 3.1.4. Measurements

The measurements were taken before and after machining procedures, and it should be stated that due to the complex outer structure of the specimens, the results were averaged as if the specimens were perfect cuboids. Measurements of the specimens designed for flexural tests were taken, with an accuracy down to 1 mm, and measurements of the specimens designed for compression tests were carried out with an accuracy of 0.1 mm. Averaged results of the measured dimensions were used for later calculations of other parameters. The length of the specimens is marked as “L”, the specimens’ width is marked as “W”, and the specimens’ height is marked as “H”.

The specimens were weighed in natural, dry, and fully saturated states to obtain values of density in these states and also to measure the material’s water absorption. The density was measured according to the standard PN-EN 12390-7, [43], and water absorption was measured according to PN-B-6250, [44]. Mass measurements were taken with an accuracy of 0.5 g. It should be stated that it is more than enough to use this level of accuracy with specimens of such size. A Radwag WLC 30/C1/R scale with a maximum measuring range of 30 kg ± 0.5 g was used for mass measurements.

The obtained measurements of the specimens before and after machining were used for further calculations. Measurements of the specimens before machining were used for calculations of density and water absorption.

#### 3.1.5. Freeze–Thaw Resistance

A special freeze–thaw chamber was used to later gather information about the frost resistance of the material used for the specimens. A freeze–thaw test was executed according to standard PN-B 6250, which was withdrawn but due to some procedures not covered by new standards, it is still in use. The specimens were fully saturated with water before being placed inside the freezing chamber and were then subjected to 25 full cycles of freezing and thawing. Each cycle consisted of four steps: saturation with water, draining, freezing, and thawing. The freeze–thaw cycle is shown in Figure 3. The specimens had to be removed before saturation with water or at the end of thawing, just before the new cycle begins so that they would not be subjected to thermal shock, which can damage them and therefore give incorrect information in mechanical tests. After being subjected to 25 freeze–thaw cycles, the specimens were analysed for weight loss, and their appearance was visually analysed for any defects such as the loss of material, cracking, or layer separation. All of the specimens were later brought to their natural moisture state and tested with the rest of the specimens designated for mechanical tests.

#### 3.1.6. Mechanical Tests

An MTS model bionic servo-hydraulic test system, with the highest attainable force of 25 kN, was used for conducting flexural and compression tests on the specimens tested for frost resistance and the specimens not exposed to unfavourable temperature changes. A Model 647.02B-22, 647 Hydraulic Wedge Grip was used in the machine for fixture placement and movement. An MTS model 810 Material Test System, with the highest attainable force of 100 kN, was used for the conduction of the compression test. A Model 647.10A, 647 Hydraulic Wedge Grip was used for fixture placement and movement in the MTS 810 machine. The use of a machine with higher attainable force was necessary for the specimens tested in the direction of longitudinal layer laying.

##### Three-Point Bending (3PB) Test

The specimens were mounted in the fixture, as shown in Figure 4, with the bottom supports’ distance equal to 16 cm ± 0.1 cm. The displacement speed of the 3PB test was set at 0.2 mm/min and changed to 2 mm/min after the first signs of failure in the tested specimen. These speeds enabled a more precise analysis of the material’s mechanical behaviour and a detailed description of the failure process. The tests were stopped when 50 mm of displacement was reached; this level of displacement allowed for the analysis of fibre and continuous reinforcement behaviour.

Equations (1) and (2) were used to obtain the information needed for further processing of obtained results. Equation (1) shows the formula for the averaged width of the specimen.
(1)$$b_{i}=\frac{\sum^{\;}W_{i}}{i}$$
where *b_i_* is the averaged width of the specimen, and *W_i_* is the measured width of the specimen.

Failure stress was calculated according to Equation (2).
(2)$$\sigma_{i}=\frac{M_{i}}{\frac{b_{i}h_{i}{}^{2}}{6}}$$
where *σ_i_* is the failure stress, *M_i_* is the failure momentum, and *h_i_* is the height of the specimen.

##### Uniaxial Compression Test

The specimens were mounted in the fixture, as shown in Figure 5a,b. Elements with flat surfaces were mounted in the wedge grips, and an additional joint was added on the upper surface to eliminate imperfection and achieve uniform height among the specimens.

Both types of specimens, i.e., the naturally cured sample and the sample subjected to freeze–thaw cycles, were tested under uniaxial compression. The tests were carried out in the stacking direction of layers and the direction of longitudinal layer laying. In addition, a few specimens were tested without continuous reinforcement for the analysis of the impact of net reinforcement on mechanical behaviour. The tests were conducted with an initial cross-head speed of 0.8 mm/min, which changed to 2 mm/min after the beginning of the failure. Finally, the tests were stopped when 15 mm of displacement of the cross-head was reached.

The final failure stress results were calculated according to Equation (3).
(3)$$\sigma_{i}=\frac{F_{i}}{A_{c_{i}}}$$
where *σ_i_* is the failure stress, *F_i_* is the failure force, and *A*_c_ is the cross-section area (*A_H_* or *A_L_*).

The cross-section areas used for stress calculations in different testing directions were calculated using Equation (4) for the specimens compressed in the height direction and Equation (5) for the specimens compressed in their length direction.
(4)$$A_{H_{i}}=W_{avg_{i}}\cdot L_{avg_{i}}$$
where *A_Hi_* is the cross-section of the specimens compressed in the height direction.
(5)$$A_{L_{i}}=H_{avg_{i}}\cdot W_{avg_{i}}$$
where *A_Li_* is the cross-section of the specimens compressed in the length direction.

## 4. Results and Discussion

### 4.1. Measurements, Observations, and Calculations

Table 4 collects the physical properties of the specimens measured before machining.

The data listed in the above tables allow for defining certain characteristics of the material used in the chosen 3DP technology. The material was characterised by high water absorption, which is probably related to the use of granulated natural cork as lightweight aggregate. There were no significant differences between the amount of water absorbed by specimens with and without continuous reinforcement (WCR), so continuous reinforcement was not correlated with high water absorption and did not have a negative impact on this property. If continuous reinforcement had any effect, it would be visible as higher or lower water absorption values, but due to the lack of changes, one can assume a lack of correlation.

Although identical printing methods and materials were used for specimen production, there were slight differences between the specimens’ dimensions and structure. The use of continuous reinforcement during specimen production facilitates the form retention of each layer, highly improves the material’s buildability, and reduces the so-called elephant’s foot effect of lower layers. The changed dimension values of the specimens after their machining for the 3PB test were later used for calculations. Additional information about the use of continuous reinforcement and its impact on the quality of the interlayer connection was gathered during the process of destruction. No voids or air pockets were found between the net or near the net’s structure, and layers were firmly connected.

### 4.2. Freeze–Thaw Test

There were no visible changes in the mass and geometrical aspects of the specimens. Further mechanical tests were conducted to measure the eventual changes in the specimens.

### 4.3. Three-Point Bending (3PB) Test

The test results and the calculated values are shown in Table 5. The specimens tested for frost resistance are marked by the letter R. The material tested for frost resistance showed a 1.0% loss of strength in the 3PB tests.

Figure 6 shows a comparison of the loading graphs plotted for the early stage of the 3PB tests of the specimens with continuous reinforcement cured at room temperature. Figure 7 shows a comparison of the loading graphs plotted for the early stage of the tests using the specimens with continuous reinforcement and 25 freeze–thaw cycles.

Both types of specimens showed similar behaviour throughout the test. At the beginning of the test, as displacement increased, the applied force increased until reaching a critical point. The critical point indicated the force of failure, which was the maximum force the specimen could withstand. After that critical point, there was a fast drop in the applied force to about 50% of the critical point value, which did not change for a while. The lack of change, reflected in the plateau of the graph, changed after another smaller peak, which led to lower forces with the tendency of aiming at 0 N.

The shape of destruction, which is probably related to the shape of the tested specimens, imperfections of their interlayers, and the distribution of cork granules in the volume of the specimens, is shown in Figure 8. The relatively low final strength, considering the high amount of cement used in the mixture, might be because the material was not compacted during the process of extrusion, as well as due to the addition of cork, which worked as the weak point in the structure [29,30,45].

### 4.4. Uniaxial Compression Test

Table 6 shows the test results and the calculated values for the specimens tested under uniaxial compression. The specimens after 25 freeze–thaw cycles are marked by “M”, and the specimens with the net reinforcement are marked by “Z”. “H” indicates that the specimen was tested along the axis of the specimen’s height, and the specimens tested along the axis of their length are marked by “L”. Four additional specimens without net reinforcement were tested to obtain information about how the applied reinforcement worked with the used materials.

There was a marked difference between the obtained results of the specimens tested in different directions. The specimens tested in the direction of the height axis showed a quick peak in the applied force, which later rapidly decreased to the smaller residual values. The specimens compressed in the direction of the length axis showed similarities to the specimens tested under the 3PB test. After reaching the point of failure, these specimens maintained a certain amount of strength, which continued until reaching another peak, followed by a rapid decrease in values. The specimens tested after 25 freeze–thaw cycles had similar tendencies in both directions of compression. All of the graphs show the early stage of loading. In addition to the changes observed in the graphs, it should also be noted that the obtained maximum values varied relative to different directions and aspects of the specimens.

Figure 9 and Figure 10 show a comparison of the graphs plotted for the specimens with net reinforcement and without additional testing for frost resistance. For the specimens without additional heat treatment, a typical difference in the obtained strength value in different compression directions was 42.6%. Similar to the results acquired during the 3PB test, the relatively low final strength might be related to the lack of compact material and the addition of cork.

Figure 11 and Figure 12 show a comparison of the graphs obtained from the tests of the specimens without net reinforcement. Differences of 28.8% were observed between the results obtained in different directions of testing. In comparison to the standard specimens reinforced with the net, the specimens without net reinforcement on average obtained 38.5% higher strength values for compression along the height direction and 23.8% higher values for compression along the length direction.

Figure 13 and Figure 14 compare the graphs obtained from the tests using the specimens with net reinforcement that also underwent 25 freeze–thaw cycles. After 25 freeze–thaw cycles, a difference in value of 42.9% was observed. The specimens treated with 25 freeze–thaw cycles showed 11.2% higher strength values in the height direction and 11.8% higher values in the length direction than the specimens without heat treatment. The counterintuitive behaviour of the composite might be related to differences in the internal structure of the specimens, which could not be observed during external visual tests. Moreover, the authors hypothesise that the changes in the internal structure caused by the formation of microcracks in the material during multiple cycles of freezing and thawing might have led to better energy transfer between the layers. Thus, the increase in strength during the uniaxial compression test may be associated with higher energy dispersion. The change in behaviour seen during the 3PB test was related to a different type of destruction process.

Figure 15a,b show different final failure modes of the specimens relative to different orientations of the layers (H—horizontal and L—longitudinal) under the compression force. In the case of the specimens tested in the direction of their height (Figure 15a), chipping and cracking were distributed over the whole composite volume, through all layers. The destruction of the specimens compressed in the direction of the specimens’ length (Figure 15b) mainly consisted of the splitting of layers and additional horizontal cracking, enhancing this separation process.

Figure 16 shows a comparison of the obtained averaged values of failure stress with standard deviations for different series of specimens and tests. For easier data reading, the series tested for compression are marked with the letter “C”, and the series used for the three-point flexural test are marked with “FLEX”. All other markings are similar to those used in Section 4.3 and Section 4.4.

## 5. Conclusions

In this study, 3D-printed beams and cubes were manufactured with printable cement-based material containing granulated cork using novel technology. Two types of reinforcement were additionally used: (1) polypropylene fibres (12 mm length) for CP layers and (2) continuous polyethylene net (7 × 7 mm) at interfaces. The specimens were tested under three-point bending and uniaxial compression along different directions of the 3D-printed composites to observe the behaviour of the printed layers and their interface connections. Part of the beams was used for testing the frost resistance of the composite.

The conducted tests allowed us to assess the physical properties and mechanical behaviour of the new 3D-printed materials. The following conclusions were drawn:The material’s water absorption by weight was 14%.The analysed composite showed adequate frost resistance. The specimens did not show any changes after freeze–thaw treatment.After 25 freeze–thaw cycles, the material showed 11.2% higher compressive toughness in the direction of the height axis and 11.8% higher values in the direction of the length axis but also showed 1.0% lower flexural strength.The use of net-type continuous reinforcement allowed for better buildability and additionally lowered slumping and elephant’s foot effects.The use of a polymer net as continuous reinforcement led to a decrease in compressive toughness, lowering it on average by 38.5% for the stacking direction and 23.8% perpendicular to the stacking direction. The net was not stiff enough, and thus it did not work as proper reinforcement. Nevertheless, it allowed for the composite to partially work after the first signs of failure because of its residual strength.The composite showed orthotropic properties. The uniaxial compressive strength in the direction of layer stacking was lower than that of the direction perpendicular to it, by 42.6% in the case of net reinforcement without heat treatment, 42.9% in the case of net reinforcement followed by the freeze–thaw test, and 29.8% without net reinforcement and heat treatment [15,41,46].

This study will be extended to the numerical analysis of mechanical response, including the complex damage mechanism and cracking process of deformed specimens. To simulate these advanced damage states, several models that are presented in [47,48,49,50,51] will be adopted.

## Figures and Tables

**Figure 1 materials-16-04235-f001:**
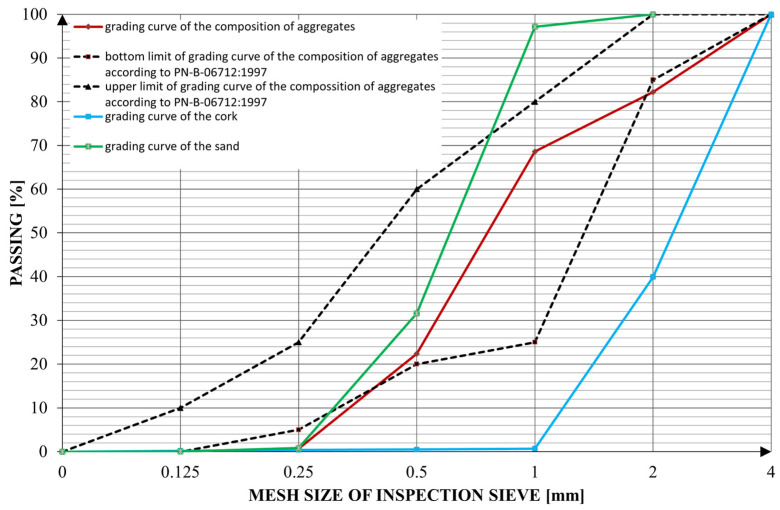
Grading curve of aggregates and their composition.

**Figure 2 materials-16-04235-f002:**
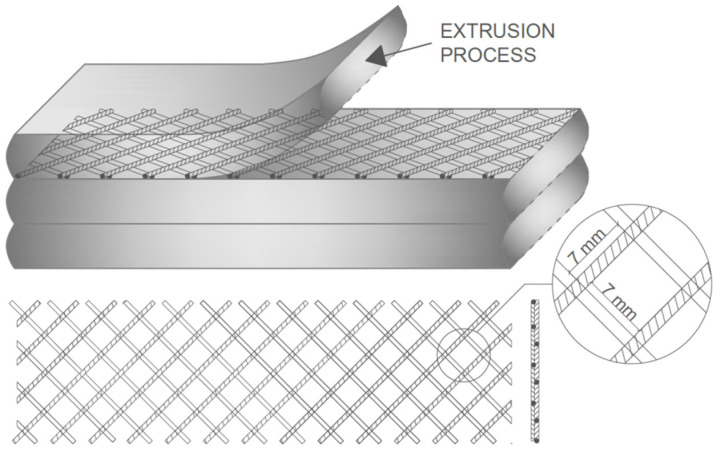
Visualisation of the printing process and net placement.

**Figure 3 materials-16-04235-f003:**
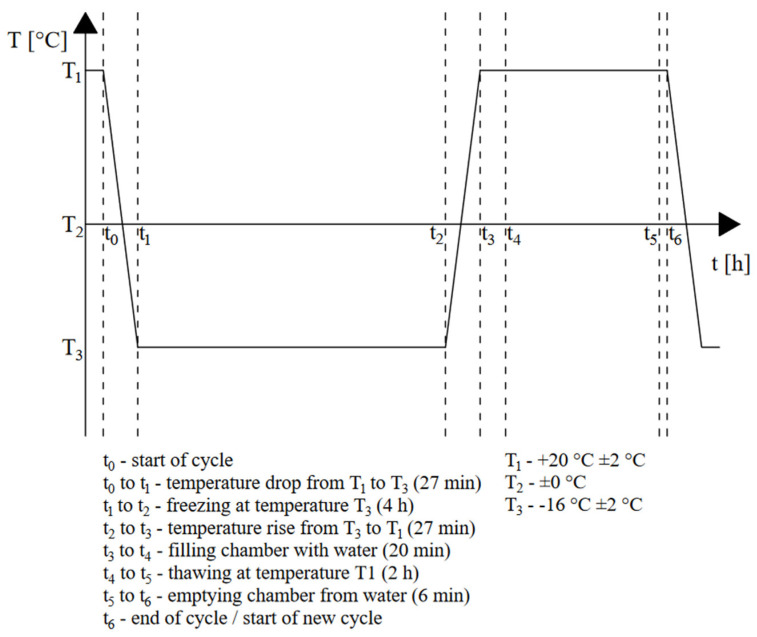
Freeze–thaw cycle.

**Figure 4 materials-16-04235-f004:**
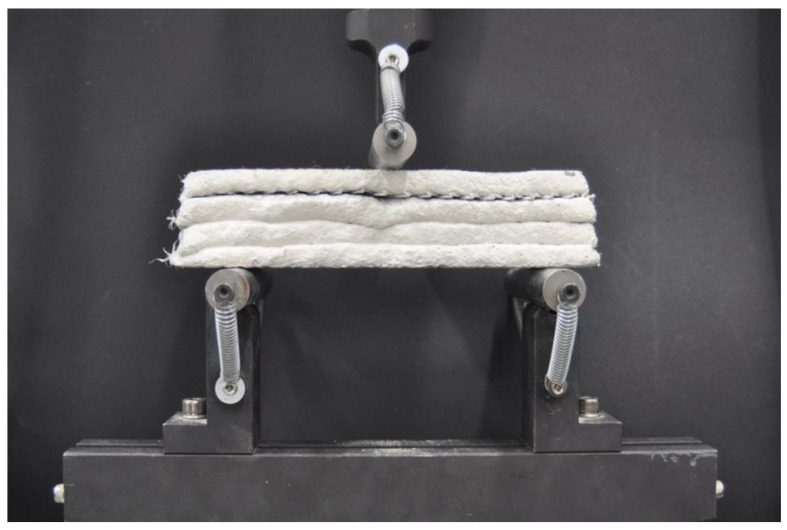
The test stand for the three-point flexural test.

**Figure 5 materials-16-04235-f005:**
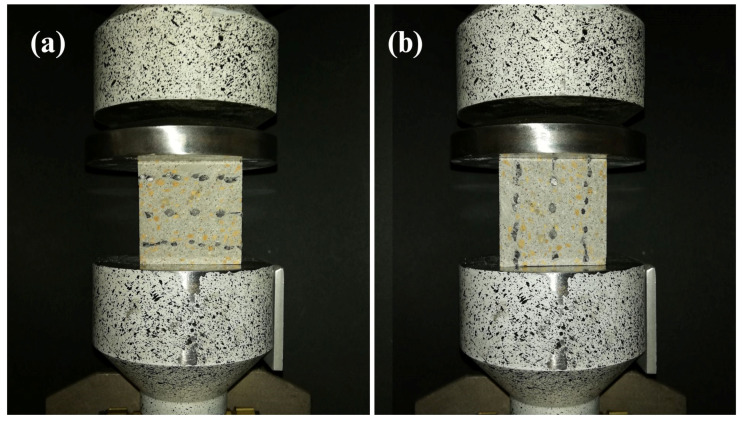
Test stand for uniaxial compression test: (**a**) specimen loaded in layer-stacking direction; (**b**) specimen loaded perpendicularly to layer-stacking direction.

**Figure 6 materials-16-04235-f006:**
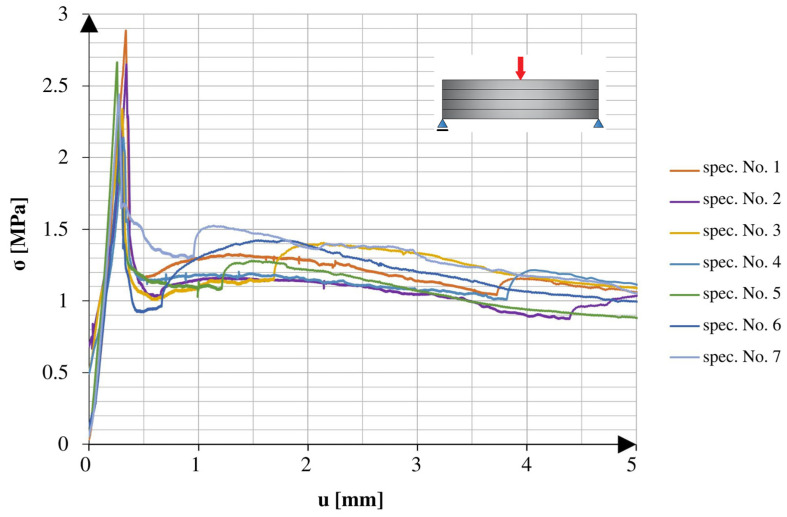
Stress–cross-head displacement relations in the 3PB test for standard specimens.

**Figure 7 materials-16-04235-f007:**
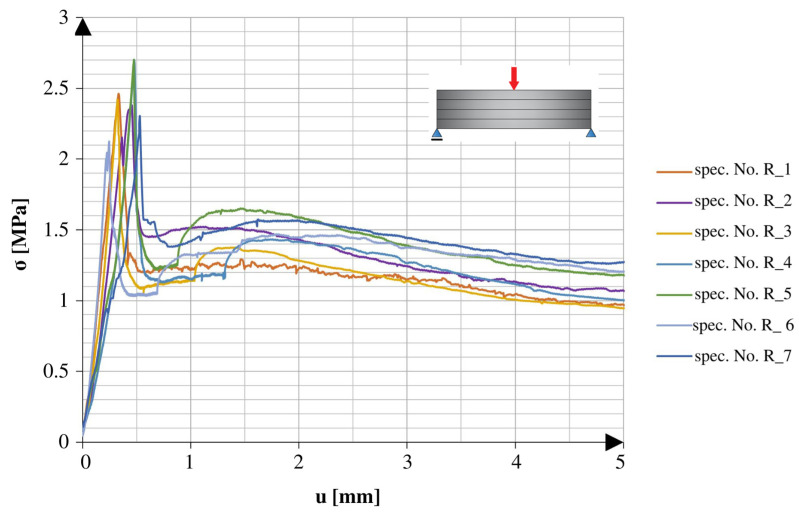
Stress–cross-head displacement relations in the 3PB test for specimens after 25 freeze–thaw cycles.

**Figure 8 materials-16-04235-f008:**
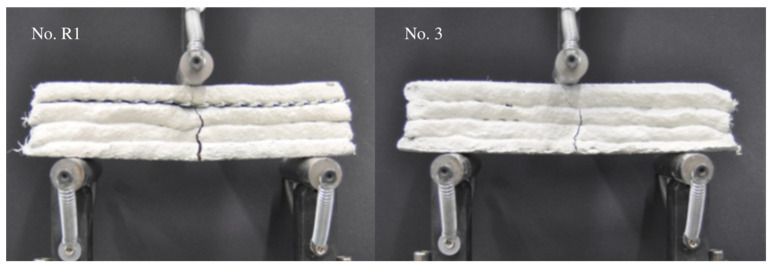
Destruction of specimen No. R1 (**left**) and No. 3 (**right**).

**Figure 9 materials-16-04235-f009:**
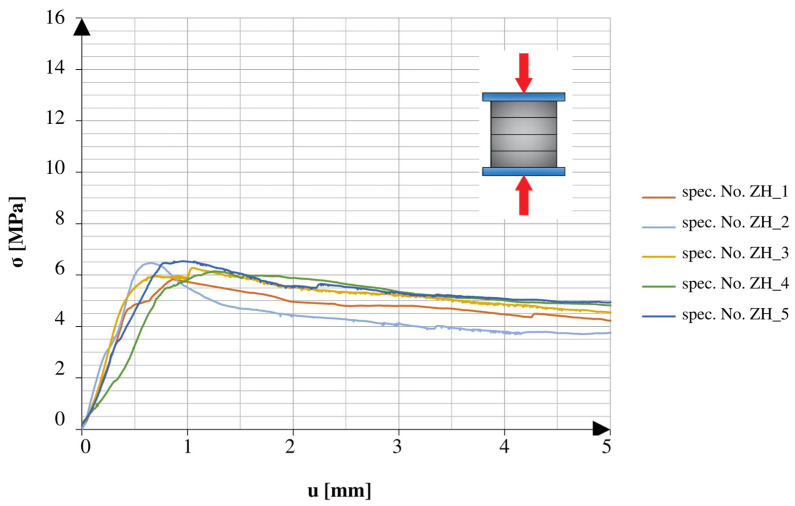
Stress–cross-head displacement relations in compression test for standard specimens with continuous reinforcement. Force was applied along the layer-stacking direction.

**Figure 10 materials-16-04235-f010:**
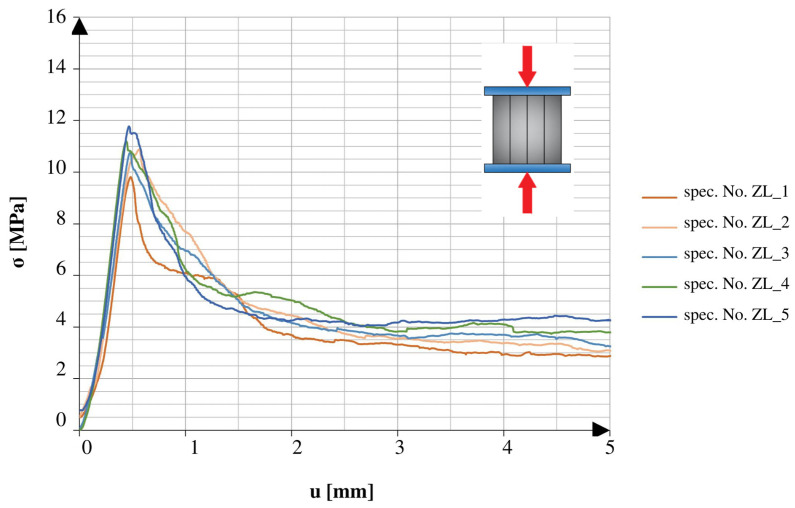
Stress–cross-head displacement relations in compression test for standard specimens with continuous reinforcement. Force was applied perpendicular to the layer-stacking direction.

**Figure 11 materials-16-04235-f011:**
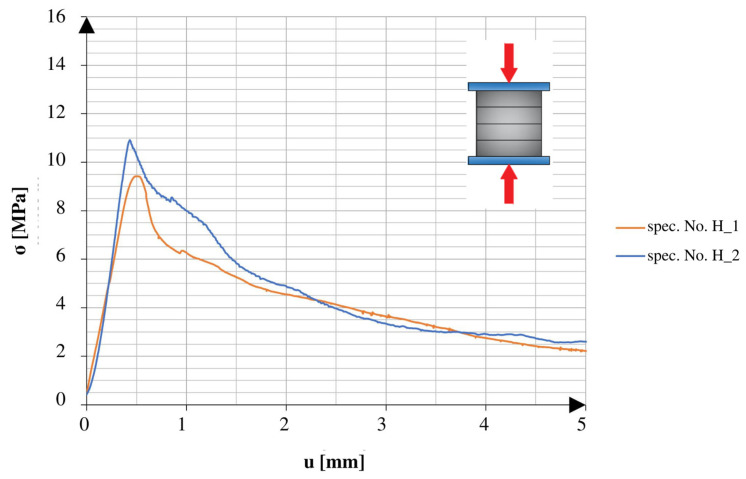
Stress–cross-head displacement relations in compression test for specimens without continuous reinforcement. Force was applied along the layer-stacking direction.

**Figure 12 materials-16-04235-f012:**
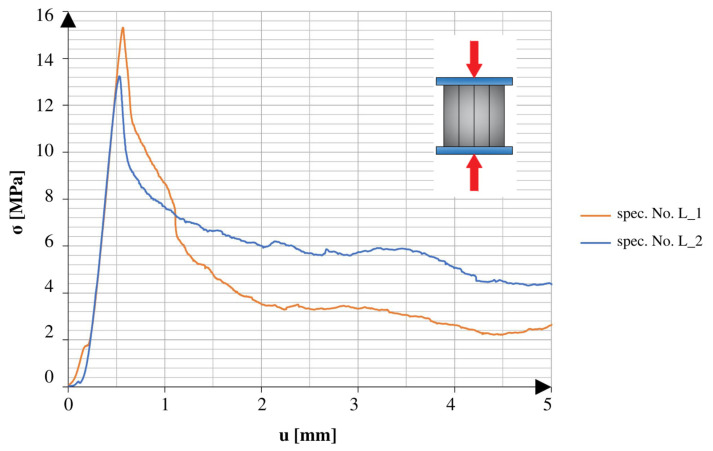
Stress–cross-head displacement relations in compression test for specimens without continuous reinforcement. Force was applied perpendicular to the layer-stacking direction.

**Figure 13 materials-16-04235-f013:**
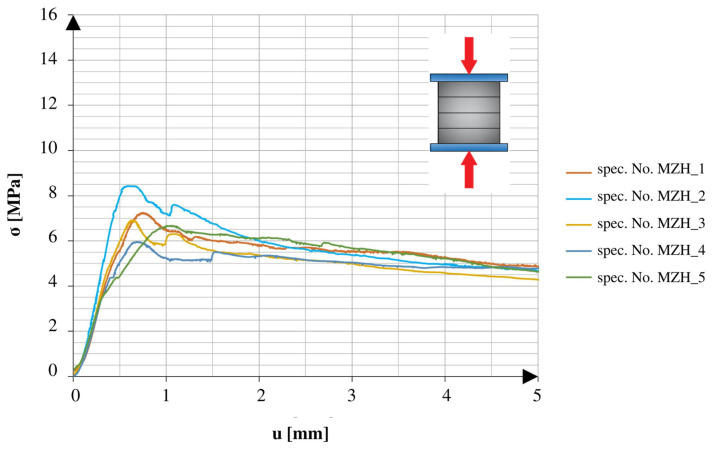
Stress–cross-head displacement relations in compression test for specimens with continuous reinforcement, after 25 freeze–thaw cycles. Force was applied along the layer-stacking direction.

**Figure 14 materials-16-04235-f014:**
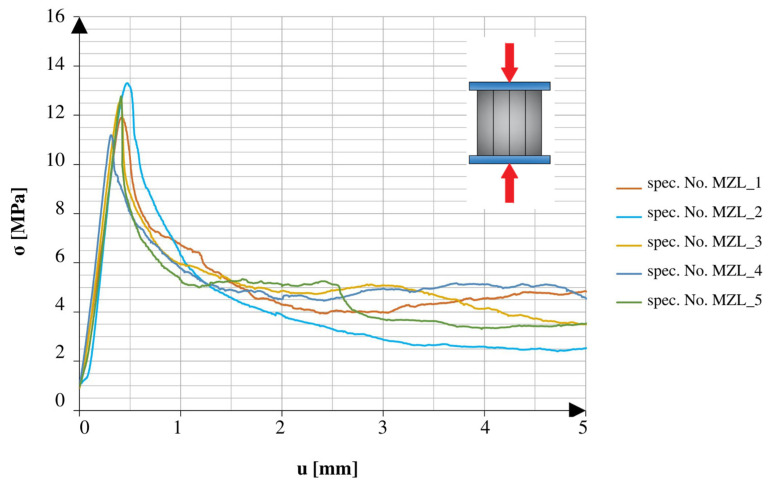
Stress–cross-head displacement relations in compression test for specimens with continuous reinforcement, after 25 freeze–thaw cycles. Force was applied perpendicular to the layer-stacking direction.

**Figure 15 materials-16-04235-f015:**
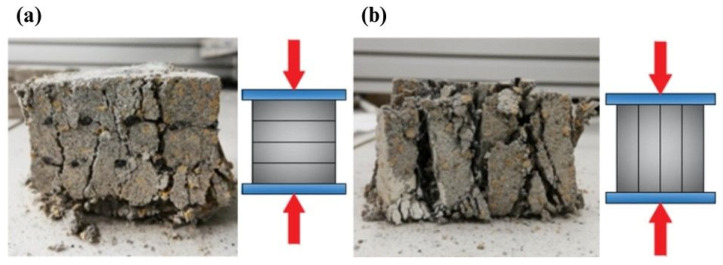
Specimen No. MZH2 (**a**; **left**) and specimen No. MZL4 (**b**; **right**) after 15 mm of displacement.

**Figure 16 materials-16-04235-f016:**
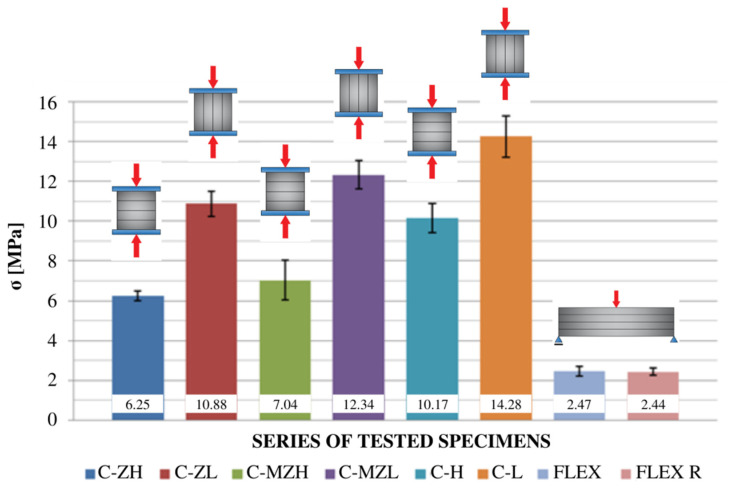
Comparison of the obtained data.

**Table 1 materials-16-04235-t001:** Grading curve of aggregates and their composition.

The Mesh Size of the Inspection Sieve (mm)	Passing (%)		
	Sand	Cork	Composition
4.0	-	100.00	100.00
2.0	100.00	39.93	82.20
1.0	97.16	0.70	68.58
0.5	31.59	0.49	22.37
0.25	0.82	0.41	0.70
0.125	0.04	0.20	0.08
0.0	0.00	0.00	0.00

**Table 2 materials-16-04235-t002:** Properties of admixture declared by the manufacturer.

Characteristics	Properties
Chloride ion content	≤0.5%
Alkaline content	≤0.8%
Aeration	Amount of air after mixing. Summarised amount of air A_1_ = (17 ± 3)% of volume
	Amount of air after 1 h of holdup ≥ A_1_ − 3%
	Amount of air with longer mixing times ≤ A_1_ + 5% and ≥ A_1_ − 5%
Corrosive influence	Contains ingredients from EN 934-1:2008 annex A. 1. only
Compressive strength	Tested mix ≥ 70% of control mix

**Table 3 materials-16-04235-t003:** Final mixture composition used for specimen preparation.

Materials	The Amount for 1 m^3^ of Mixture
Cement	381.2 kg
Sand	747.9 kg
Cork granules	13.6 kg
Water	181.8 kg
Admixture	1.5 kg
Fibre reinforcement	1.4 kg

**Table 4 materials-16-04235-t004:** Specimens’ properties.

Specimen No.	Density			Water Absorption	
	ρ (g/cm^3^)	ρ_s_ (g/cm^3^)	ρ_n_ (g/cm^3^)	n_w_ (%)	n_o_ (%)
R1	1.635	1.611	1.805	12%	19%
R2	1.612	1.590	1.783	12%	19%
R3	1.374	1.353	1.534	13%	18%
R4	1.606	1.580	1.794	14%	22%
R5	1.502	1.477	1.678	14%	20%
R6	1.620	1.592	1.808	14%	22%
R7	1.587	1.560	1.775	14%	22%
1	1.486	1.462	1.666	14%	20%
2	1.600	1.572	1.782	13%	21%
3	1.643	1.617	1.837	14%	22%
4	1.466	1.443	1.640	14%	20%
5	1.452	1.431	1.630	14%	20%
6	1.486	1.463	1.664	14%	20%
7	1.544	1.509	1.717	14%	21%
WCR1	1.671	1.646	1.857	13%	21%
WCR2	1.609	1.586	1.789	13%	20%
Avarage	1.556 ± 0.082	1.531 ± 0.081	1.735 ± 0.087	14% ± 1%	20% ± 1%

**Table 5 materials-16-04235-t005:** Test results and calculations.

Specimen No.	Failure Stress	Specimen No.	Failure Stress
	σ (MPa)		σ (MPa)
R1	2.462	1	2.888
R2	2.383	2	2.651
R3	2.422	3	2.339
R4	2.700	4	2.140
R5	2.703	5	2.665
R6	2.125	6	2.140
R7	2.301	7	2.442
Avarage	2.442 ± 0.193	Avarage	2.466 ± 0.262

**Table 6 materials-16-04235-t006:** Test results and calculations.

Specimen No.	Failure Stress	Specimen No.	Failure Stress
	σ (MPa)		σ (MPa)
ZH1	5.835	ZL1	9.816
ZH2	6.468	ZL2	10.877
ZH3	6.280	ZL3	10.760
ZH4	6.145	ZL4	11.194
ZH5	6.536	ZL5	11.775
Avarage	6.253 ± 0.250	Avarage	10.884 ± 0.640
MZH1	7.242	MZL1	11.905
MZH2	8.430	MZL2	13.308
MZH3	6.912	MZL3	12.538
MZH4	5.959	MZL4	11.195
MZH5	6.671	MZL5	12.773
Avarage	7.043 ± 0.812	Avarage	12.344 ± 0.730
H1	9.423	L1	15.321
H2	10.917	L2	13.243
Avarage	10.170 ± 0.747	Avarage	14.282 ± 1.039

## Data Availability

The data presented in this study are available on request from the corresponding author.

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
