# Peer review of "A 3D Printing Method of Cement-Based FGM Composites Containing Granulated Cork, Polypropylene Fibres, and a Polyethylene Net Interlayer"

_materials, 2023, doi:10.3390/ma16124235_

Round 1
Reviewer 1 Report
The paper titled “A novel method of 3D printing of cement-based FGM composites containing granulated cork, polypropylene fibres and polyethylene net interlayer.” is a great paper to be published in the materials journal, however, many changes are required to be done prior to its publication. First in title should not have dot at the end of the title, and can be improved the content to represent a better view of the paper’s content. The title should be renamed, please remove the novel.
Abstract
The authors should show some results at the end of the paper.
Introduction
Authors should read all earlier papers regarding 3dp concrete with content of fiber. There are many studies authors did not read on them such as “Effects of deposition velocity in the presence/absence of E6-glass fibre on extrusion-based 3D printed mortar”, also authors should tell the truth these materials are mortar not concrete.
Authors should read first review papers that content bunch of works “Large-Scale 3D Printing for Construction Application by Means of Robotic Arm and Gantry 3D Printer: A Review” and also all related to the reinforcement of concrete with fibers. Please read them to have a better representation in the introduction.
Also authors should be aware to use whether additive manufacturing or 3DP.
Line 50 have a bunch of citation please take them out and cite them one by one and tell which study work on what, or remove some of them. Similar for line 72.
Materials
Figure 2.1 is too small in value and text please enlarge them and use better font.
Figure 3.2 it is just repeat of other paper please change and show something new
Figure 3.4 the quality of printing so low and there is a huge gap between layers why?
Figure 3.5 change to better image.
Figure 4.1 and figure 4.2 it is not clear whether the results are okay or correct. Please use other software and high resolutions to see all details by readers.
Figure 4.3 this is not sure that the checking strength of specimens are okay because some of them have huge gaps.
For figure 4.8 and figure 4.9 these results seem not to be okay because the strength on figure 4.8 must be higher than the figure 4.9. please check again.
Conclusion
Please highlight all outcomes by bullet points with clear shows of outcomes.
References:
This is required all necessary earlier studies and reviews to be included, this information from the list is not enough.
It can be improved to make the reader easy to flow in reading the manuscript.
Author Response
REVIEWER 1
Thank you for your time and valuable comments. Please find our answer.
The paper titled “A novel method of 3D printing of cement-based FGM composites containing granulated cork, polypropylene fibres and polyethylene net interlayer.” is a great paper to be published in the materials journal, however, many changes are required to be done prior to its publication. First in the title should not have a dot at the end of the title, and can be improved the content to represent a better view of the paper’s content. The title should be renamed, please remove the novel.
Answer: "Novel" and "dot" were removed
Abstract
The authors should show some results at the end of the paper.
Answer: The missing elements of the abstract have been supplemented.
Introduction
Authors should read all earlier papers regarding 3dp concrete with the content of fiber. There are many studies authors did not read, such as “Effects of deposition velocity in the presence/absence of E6-glass fibre on extrusion-based 3D printed mortar”, (also authors should tell the truth these materials are mortar not concrete.
Answer: This is stated in line 71 to 75, authors never called their material a concrete. Line 95, “3D printable composite material” was used for naming the proposed material. Line 204, name “concrete” was changed to “mortar”. In this case the word “concrete” was being used as an example not the name of used material. It was finally changed according to proposition of the reviewer. The mentioned studies were read and taken into consideration to improve this text.
Authors should read the first review papers that contain a bunch of works “Large-Scale 3D Printing for Construction Application by Means of Robotic Arm and Gantry 3D Printer: A Review” and also all related to the reinforcement of concrete with fibers. Please read them to have a better representation in the introduction.
Answer: Thank You for this remark especially. The review paper improved of our point of view. The text was improved by knowledge taken from them.
Also, authors should be aware to use whether additive manufacturing or 3DP.
Answer: The nomenclature has been unified in whole paper.
Line 50 have a bunch of citation please take them out and cite them one by one and tell which study work on what, or remove some of them. Similar to line 72.
Answer: The citation were taken out of bunches and spread across the text. The citations which strongly cover the described phenomenon's are left, the rest were removed.
Material
Figure 2.1 is too small in value and text please enlarge them and use a better font.
Answer: The quality of Fig 2.1 was improved.
Figure 3.2 it is just a repeat of the other paper please change and show something new
Answer: The goal of this figure was to show the dimension measurement scheme of particular specimens. In a specific journal, it is required to preserve the possibility of full reproducibility of scientific research. According to your remarks, this figure was deleted. The necessary information is included in the text.
Figure 3.4 the quality of printing is so low and there is a huge gap between layers why?
Answer: The improvement of printing quality is steel subject of our work. The addition of net reinforcement changes the behaviour of the green mortar mixture. The visible gap is a result of the net stopping the material from overflowing on the sides. This gap has a surface character and is not present in the cross-section of the specimen. The effect of such gaps on the derived results is eliminated by multi-point measurement and averaging the specimen’s dimensions. Observation of damaged and machined cross sections shows good connection between layers (see Fig 3.4).
Figure 3.5 change to better image.
Answer: The image was replaced
Figure 4.1 and figure 4.2 it is not clear whether the results are okay or correct. Please use other software and high resolutions to see all details by readers.
Answer: The quality of figures: 4.1, 4.2, 4.4 to 4.9 was improved
Figure 4.3 this is not sure that the checking strength of specimens are okay because some of them have huge gaps.
Answer: In Fig 4.3 the dark colour of the inter-layer net causes of enhancing the gap graphic illusion. Generally in this study, the specimens were machined in the top and bottom parts to achieve flat and parallel surfaces required in laboratory tests. In three-point bending tests, the non-loaded surfaces have been left in a natural state. All gaps and shape irregularities are taken into account by averaging the thickness of the considered beam. The all of gaps have a surface character that is visible in the figures which present the machined cross-section of exemplary specimens like Figure 3.4.
For figure 4.8 and figure 4.9 these results seem not to be okay because the strength on figure 4.8 must be higher than the figure 4.9. please check again.
Answer: These relations are covered by literature [15], [40], [42]. In this case, are dealing with the 2-phase layered composite. The phases can be distinguished here by locally equivalent mechanical properties. The strong and stiff phase is mortar while the interlayer phase is weak and porous. Therefore in case of force applied in the stacking direction, the interlayer phase dominates the response of material while the force is applied in a perpendicular direction the material response is dominated by the strongest mortar phase. The mathematical explanation of some phenomena will be the subject of our future papers.
Conclusion
Please highlight all outcomes by bullet points with clear shows of outcomes.
Answer: Outcomes changed to bullet points
References:
This is required all necessary earlier studies and reviews to be included, this information from the list is not enough.
Answer: The list has been expanded for the most valuable papers for this study

Reviewer 2 Report
General review:
Overall, the authors tried to propose the use of novel 3D printing cement-based composite with natural, granulated cork used as lightweight aggregate and additional reinforcement by (1) continuous polyethylene interlayer net combined with the use of (2) polypropylene fibre reinforcement. Assessment of used materials’ multiple physical and mechanical properties during the 3D printing process and after curing, verifies the applicability of the new composite in additive manufacturing. The composite exhibits orthotropic properties. The uniaxial compressive strength in the direction of layer stacking is lower than that of the direction perpendicular to it. Use of the polymer net as a continuous reinforcement led to decreasing in compressive toughness. However, the net reinforcement additionally lowers slumping and elephant foot effects. Moreover, the net reinforcement introduces residual strength which allows for further work of the composite after failure of the brittle material. Data obtained during the process can be used for further development and improvement of 3D printable building materials. However, before it can be published, I have some questions about this article and some suggestions:
Minor review:
1. Several sentences starting in Abstract and Introduction were too general to distinguish this paper. Please start directly to mention the main issues in this paper.
2. The main idea is tight bonding between the reinforcement and matrix. You should comment these.
3. In the experimental section, you should more mention the machines and instruments explained in Results and discussion in more detail.
4. With regard to 3D printing, you should read and cite this paper, “Additive manufacturing of a shift block via laser powder bed fusion: the simultaneous utilisation of optimised topology and a lattice structure”.
5. Some of the references are too old and thus replace them.
Author Response
REVIEWER 2
Thank you for your time and valuable comments. Please find our answer.
Minor review:
- Several sentences starting in Abstract and Introduction were too general to distinguish this paper. Please start directly to mention the main issues in this paper.
Answer: The general sentences were rewritten to highlight the main issues in this paper.
- The main idea is tight bonding between the reinforcement and matrix. You should comment these.
Answer: In this study, we describe the effect of using the reinforcement for the macro-scale response of the material to applied loadings. In our feeling, the comment on the tightening of the bonding between the reinforcement and matrix requires another study. I.e. the special treatment of reinforcement effect should be considered. We are concerning such studies in future investigations.
- In the experimental section, you should more mention the machines and instruments explained in Results and discussion in more detail.
Answer: The text was overviewed. The crucial equipment is mentioned in the section "Methods" There were carefully checked so that machines not described before do not appear in the "Results and Discussion" section.
- With regard to 3D printing, you should read and cite this paper, “Additive manufacturing of a shift block via laser powder bed fusion: the simultaneous utilisation of optimised topology and a lattice structure”.
Answer: Thank You for this remark especially. The pointed paper is valuable for our team. It was carefully read. We cite this paper in this study.
- Some of the references are too old and thus replace them.
Answer: Most of the too-old references were deleted or replaced. The most important references for this study were kept despite the old publication date.

Round 2
Reviewer 1 Report
The paper is now in better version only minor grammatical issues, it must revise by authors
It is okay but is not in high level.